# Limitations in predicting reduced susceptibility to third generation cephalosporins in *Escherichia coli* based on whole genome sequence data

Anna Heydecke[1], Hong Yin[2], Eva Tano[3], Susanne Sütterlin[4]*

**1** Center for Research and Development Gävleborg, Uppsala University, Gävle, Sweden, **2** Department of Clinical Microbiology, Falun Hospital, Falun, Sweden, **3** Department of Medical Sciences, Uppsala University, Uppsala, Sweden, **4** Department of Women's and Child's Health, International Maternal and Child Health, Uppsala University, Uppsala, Sweden

* susanne.sutterlin@kbh.uu.se

## Abstract

Prediction of antibiotic resistance from whole genome sequence (WGS) data has been proposed. However, the performance of WGS data analysis for this matter may be influenced by the resistance mechanism's biology. This study compared traditional antimicrobial susceptibility testing with whole genome sequencing for identification of extended-spectrum beta-lactamases (ESBL) in a collection of 419 *Escherichia coli* isolates. BLASTn-based prediction and read mapping with srst2 gave matching results, and in 381/419 (91%) isolates WGS was congruent with phenotypic testing. Incongruent results were grouped by potential explanations into biological-related and sequence analysis-related results. Biological-related explanations included weak ESBL-enzyme activity (n = 4), inconclusive phenotypic ESBL-testing (n = 4), potential loss of plasmid during subculturing (n = 7), and other resistance mechanisms than ESBL-enzymes (n = 2). Sequence analysis-related explanations were cut-off dependency for read depth (n = 5), too stringent (n = 3) and too loose cut-off for nucleotide identity and coverage (n = 13), respectively. The results reveal limitations of both traditional antibiotic susceptibility testing and sequence-based resistance prediction and highlight the need for evidence-based standards in sequence analysis.

**Data Availability Statement:** All 419 files are available from http://enterobase.warwick.ac.uk/ or ENA/SRA/DDBJ databases (accession number(s) ESC_OA4486AA to ESC_OA4685AA,

## Introduction

Rapid and correct determination of both resistance and susceptibility to antibiotics is regarded as an essential part in combating antimicrobial resistance [1]. Today, standardized phenotypic antimicrobial susceptibility testing combined with clinical breakpoints, is the standard method for reporting treatment options to clinicians. These methods determine the potential of antibiotics to inhibit bacterial growth at varying concentration levels and relate these results to clinical treatment outcome by clinical breakpoints. Minimal inhibition concentration (MIC) has a central role in deciding the choice of antibiotic drugs and dosage regime [2]. While prediction

ESC_WA3873AA to ESC_WA3893AA,
ESC_MA4674AA to ESC_MA4690AA and project
number PRJEB17631).

**Funding:** SS holds a grant from ALF funds (ALF
810901) and the Swedish Research Council (2019-
05909), and AH holds a grant from the Center for
Research and Development Gävleborg (CFUG-
698631). Generation of sequence data was
supported by a grant from Afa Insurance, Sweden
(grant number 150411) and Alf-de-Ruvo memorial
foundation, Sweden. The funders had no role in
study design, data collection and analysis, decision
to publish, or preparation of the manuscript.

**Competing interests:** The authors have declared
that no competing interests exist.

of treatment success based on MIC generally works well, it fails in some cases since bacterial properties like biofilm production, gene amplification or formation of persister cells can influence the efficacy of antibiotic treatment. [3–5].

Common resistance mechanisms in *Escherichia coli* mediating reduced susceptibility to third generation cephalosporins include acquired extended- spectrum beta-lactamases (ESBL), or AmpC-beta-lactamases (AmpC); other common beta-lactamases (OXA); and reduced permeability [6–8]. Genetic methods can provide rapid assessments of antibiotic resistance markers of beta-lactamases, and they can be performed directly on the specimens. The most common molecular approach is the use of PCRs to screen for a limited number of resistance genes as part of surveillance and control of resistant bacteria, but this approach is not commonly used to guide antimicrobial therapy. A potential benefit of genetic methods is the detection of silent or lowly-expressed genes that may be induced during treatment [9–11]. Further, a variety of mechanisms can increase gene expression, leading to clinical resistance in bacteria that appear to be phenotypically susceptible based on testing of primary cultures: transcriptional induction, gene amplification or mutational events [5,10,12,13]. For bacterial species that possess inducible resistance mechanisms, genetic methods can play a complementary role in risk stratification of isolates classified as susceptible based on phenotypic methods. While most methods in clinical microbiology are subject to interlaboratory standardization, this is still lacking for bioinformatical analysis [14,15]. Here, we examined the ability of WGS to detect the reduced susceptibility to third generation cephalosporins in *E. coli* as compared to standard antimicrobial susceptibility testing.

## Material and methods

### Bacterial isolate collections

A total of 419 *E. coli* isolates, collected through different previous studies, were included in the study in order to represent a diverse clinical spectrum and geographic areas where susceptibility testing is important: 326 isolates were consecutively selected (88 isolates from urine cultures from Spain, Germany and Sweden (2016) [16], 200 uropathogenic and blood stream isolates, isolates from Sweden (2016) (http://enterobase.warwick.ac.uk/, ESC_OA4486AA to ESC_OA4685AA) and 38 neonatal blood stream isolates (2005 – 2020) (http://enterobase.warwick.ac.uk/, ESC_WA3873AA to ESC_WA3893AA, ESC_MA4674AA to ESC_MA4690AA)); and 93 isolates from ESBL/AmpC-producing isolate collections to increase the amount of isolates with resistance to third generation cephalosporines (52 ESBL-producing isolates from urine cultures from Sweden (2011 – 2016) [17], 41 ESBL-producing isolates from feces of neonates and their mothers from China (2012)(ENA/SRA/DDBJ databases under project reference PRJEB17631)). Species identification was performed using standard laboratory procedures and automated species identification systems Vitek 2® AutoMicrobic System (bioMerieux, Marcy-l'Étoile, France) or MALDI-TOF (Bruker Daltonics, Billerica, Massachusetts, USA). All isolates were stored in glycerol stock at -80°C.

### Phenotypic detection of ESBL-production

All susceptibility testing was performed according to the European Committee for Antimicrobial Susceptibility testing (EUCAST) and the Nordic Committee on Antimicrobial Susceptibility Testing (NordicAST). All isolates were phenotypically tested for susceptibility to cefpodoxime (10 μg; > = 21 mm) or cefadroxil (30 μg; > = 12 mm). When disc diffusion (Oxoid, UK) rendered an isolate not susceptible, they were further analyzed for production of ESBL by using a diffusion synergy test with clavulanic acid (10 μg) and the cephalosporins cefotaxime (5 μg), ceftazidime (10 μg) and cefepime (30 μg) [18]. Isolates were categorized as

reduced susceptibility according to clinical breakpoints: cefotaxime: < 20 mm; ceftazidime: < 22 mm; or cefepime: < 27 mm. Isolates were categorized as classical ESBL-phenotype, when synergy between clavulanic acid and the tested cephalosporins was observed. For isolates with negative synergy test but reduced susceptibility to cefotaxime or ceftazidime and cefoxitin (30 μg; < 19 mm), an AmpC-type was suspected, and thus categorized as AmpC-phenotype. Isolates that meet the criteria for both ESBL and AmpC production were assumed to produce both enzymes, and those not meeting the criteria for ESBL nor AmpC were categorized as "others".

### Whole genome sequencing based detection of ESBL genes

Bacterial genomic DNA was extracted and purified using a Wizard® Genomic DNA Purification Kit (Promega, Madison, Wisconsin, USA) according to the manufacturer's recommendations for Gram-negative bacteria with the exception that DNA was rehydrated with 10 mM Tris-HCl (pH 8.0). The quality of the extracted DNA was controlled by gel electrophoresis and spectrophotometry. DNA concentrations were measured using Quant-iT dsDNA BR assay and a Qubit instrument (Invitrogen, Waltham, Massachusetts, USA). After standardizing the DNA extracts, the samples were transferred to the Oxford Genome Center (Oxford, UK) or the National Genomics Infrastructure (Stockholm, Sweden) for library preparation and whole-genome sequencing. Fragmented DNA was end-repaired, A-tailed, adapter-ligated, and amplified using Nextera DNA library Prep (Illumina, San Diego, California, USA). Sequencing was performed on an Illumina HiSeq4000 platform or a NovaSeq SP-300 platform generating 150 bp paired-end reads.

Paired-end reads were assembled using SPAdes assembler (v3.11.1, https://cab.spbu.ru/) using the–careful flag and kmer lengths 21, 33, 55, 77, 99, 127. Species confirmation was performed using the rMLST species tool on http://pubmlst.org/rmlst, read quality was checked using FastQC (v0.11.9) and draft genome quality was examined using QUAST (v5.0.2, http://bioinf.spbau.ru/quast). Sequences were included when coverage was at least 50x and covered at least 80% of the reference genome E. coli (MG1655; Genebank accession number NC_000913). Resistance determinants were searched for by BLASTn on the draft genomes [19] and by read mapping using srst2 (0.2.0) with CARD (https://card.mcmaster.ca/; October 2020) as reference database.

BLASTn algorithm was run with standard settings, and the resulting output was parsed using the SeqIO module in Biopython [20] using variable nucleotide identity thresholds (70%, 80%, 90%, 100%) and minimum coverage (40%, 60%, 80%, 90%, 100%). Read mapping was performed using srst2 as recommended by the developers (default minimal coverage 0.9) and all outputs were considered for comparisons, regardless of read depth as the number of times each individual base had been sequenced.

### Comparison of methods and statistical methods

Comparisons of the methods were performed using the test performance parameters sensitivity and specificity. Where appropriate, statistical variance was expressed by means and standard deviations when the data were normally distributed, otherwise median with range was used.

## Results

### General comments on the dataset

Species verification on sequence data confirmed the correct species identification as *E. coli* and the purity of the whole-genome extracts. All sequences had a coverage above 50x, and covered

>80% of the reference genome *E. coli* M1655 and all contigs produced by SPAdes were included in the BLASTn analysis.

The median coverage of the high-quality short reads from all collections was 187x (range 100–739). Draft genomes produced by SPAdes resulted in a median contig number of 199 (range 34–3943) for all contigs, the median N50 value was 247,289 (range 7,878–2,444,793) and all contigs covered a median of 83% (range 80–95%) of the reference genome, the mean (SD) total length of nucleotides assembled in the draft genome was 5,186,329 (326,187) bp.

## Susceptibility testing

In this isolate collection, 30% (125/419) showed reduced susceptibility to third generation cephalosporins according to disc diffusion tests. Among these, 71% (89/125) isolates met the criteria for classical ESBL-phenotype, 22% (28/125) isolates met the criteria for AmpC-production, and 2% (2/125) isolates for a combined ESBL- and AmpC-phenotype. The remaining six isolates (5%, 6/125) showed reduced susceptibility to third generation cephalosporins; these isolates had reduced susceptibility solely for cefotaxime (FÖRL14012), cefotaxime and cefepime (FÖRL14048) or ceftazidime (B1309860, B1608125, U1310875, 18BLO055671). A total of 70% (294/419) were susceptible according to clinical breakpoint definitions.

## BLASTn search and srst2

For the BLASTn search, the number of isolates with at least one *bla* gene known to be related to ESBL/AmpC-production, varied with different combinations of thresholds for percent nucleotide identity and coverage. Thus, depending on the chosen thresholds, the number of alleles ranged from 116 (116/419; 27%) for the 100% identity and 100% coverage setting to 135 (135/419; 32%) for the 70% identity and 40% coverage setting. The read mapping-based application srst2 reported *bla*-genes related to ESBL/AmpC-production for 122 isolates (122/419; 29%) irrespective of the level of mismatches, depth or coverage, 117 isolates (117/419; 28%) when $\geq 1$ mismatches were allowed and the depth was > 10x, and 115 (115/419; 27%) perfect hits. The reported *bla* genes linked to ESBL/AmpC-production were in accordance with reported *bla* genes from BLASTn searches, and the *bla* genes belonged to $bla_{CTX-M}$ group 1 (n = 52), $bla_{CTX-M}$ group 9 (n = 46), $bla_{CMY}$ (n = 23), $bla_{DHA}$ (n = 7), $bla_{CTX-M}$ group 8 (n = 1), and $bla_{SHV}$ (n = 1); eight isolates carried ESBL-genes belonging to two different *bla* types. (Table 1).

## Comparison of methods

Overall, results for tests of phenotypic ESBL/AmpC-production and ESBL/AmpC-prediction based on BLASTn and srst2 resulted in congruent results for 91% of the isolates (381/419). When considering all tested cut-off values for BLASTn and srst2, sequence-based methods had a cumulative sensitivity and specificity to predict resistance to third-generation cephalosporins of 93% and 94%, respectively. Phenotypic methods had a cumulative sensitivity and specificity to predict *bla* genes of 86.4–89.6% and 94.2–99.3%, respectively. However, when using cut-off values of 0.8 for both coverage and identity, BLASTn predicted resistance to third generation cephalosporins with a sensitivity and specificity of 92.0% and 98.6%, respectively. High cut-off values of 100% for coverage and identity lead to decrease of sensitivity (89.6%) as relevant alleles were missing in the database at that point of time. Similar results were seen for srst2 when also including results marked uncertain (sensitivity and specificity of 92.8% and 98.0%), for results where $\geq 1$ mismatches were allowed and the depth was > 10x (sensitivity and specificity of 90.4% and 98.6%) and for perfect matches (sensitivity and specificity of 88.8% and 98.6%).

**Table 1. Beta-lactamase genes found in the isolate collections.** The overview includes all results from read mapping, BLASTn results were always congruent except for isolate U1416006.

| *bla*-type | Count | Frequency of isolates with reduced susceptibility to third generation cephalosporines carrying *bla* |
|---|---|---|
| **$bla_{CTX-M\ 1}$** | | 47/52 (90%) |
| $bla_{CTX-M-1}$ | 3 | |
| $bla_{CTX-M-15}$, $bla_{CTX-M-15-like}$ [A] | 37 | |
| $bla_{CTX-M-3}$, $bla_{CTX-M-3-like}$ | 5 | |
| $bla_{CTX-M-42}$ | 1 | |
| $bla_{CTX-M-55}$ | 6 | |
| **$bla_{CTX-M\ 8}$** | | 1/1 (100%) |
| $bla_{CTX-M-8}$ | 1 | |
| **$bla_{CTX-M\ 9}$** | | 45/46 (98%) |
| $bla_{CTX-M-14}$, $bla_{CTX-M-14-like}$ [B] | 24 | |
| $bla_{CTX-M-24}$ | 1 | |
| $bla_{CTX-M-27}$ | 11 | |
| $bla_{CTX-M-65}$ | 3 | |
| $bla_{CTX-M-9}$, $bla_{CTX-M-9-like}$ | 3 | |
| $bla_{CTX-M-132-like}$ | 3 | |
| $bla_{CTX-M-64-like}$ [C] | 1 | |
| **$bla_{SHV}$** | | 1/2 (50%) |
| $bla_{SHV-12}$ | 1 | |
| $bla_{SHV-102-like}$ | 1 | |
| **$bla_{CMY}$** | | 23/23 (100%) |
| $bla_{CMY-2}$ | 20 | |
| $bla_{CMY-15-like}$ | 1 | |
| $bla_{CMY-42}$ | 2 | |
| **$bla_{DHA}$** | | 7/7 (100%) |
| $bla_{DHA-1}$ | 7 | |
| **$bla_{TEM}$** | | 53/147 (36%) |
| $bla_{TEM-1D}$ | 147 | |
| $bla_{TEM-135}$ | 1 | |
| **$bla_{OXA}$** | | 22/27 (81%) |
| $bla_{OXA-1}$ | 26 | |
| $bla_{OXA-2}$ | 1 | |

[A] U16752: srst2: $bla_{CTX-M-15-like}$ (coverage 0.93, depth 2.2x, 61 holes), BLASTn $bla_{CTX-M-15-like}$ (identity 1.0, coverage 0.4).

[B] FÖRL14028: srst2: $bla_{CTX-M-14-like}$ (coverage 0.99, depth 3x, 1 snp,4 holes), BLASTn: $bla_{CTX-M-14-like}$ (identity 0.99, coverage 0.67); B1605999: srst2: $bla_{CTX-M-14-like}$ (coverage 0.92, depth 2.7x, 70 holes), BLASTn: $bla_{CTX-M-14-like}$ (1.0 identity, 0.78 coverage).

[C] U1416006: srst2: $bla_{CTX-M-64-like}$ (coverage 0.93, depth 53x, 19snp1indel,60holes); BLASTn: $bla_{CTX-M-132-like}$ (0.92 identity, 0.92 coverage).

Incongruent test results (9%, 38/419) could be grouped by potential explanations into biology-related and sequence analysis-related. (Table 2). Biology-related explanations included the following isolates: four isolates with consistent detection of genetic resistance that were fully

**Table 2. Summary of results from three methods (antimicrobial susceptibility testing, BLASTn and srst2) for diagnosis of ESBL/AmpC-production or ESBL/AmpC-genes on *E. coli* isolates (n = 419).**

| | N | Phenotype | Genotype | Potential explanations |
|---|---|---|---|---|
| **Biology-related** | 4 | No resistance | ESBL/AmpC-genes detected | Weak ESBL-enzyme activity or silent R-gene |
| | 4 | No ESBL-activity detected | ESBL/AmpC-genes detected | Synergy test or AmpC test failed to reveal mechanism |
| | 7 | ESBL-activity detected | No ESBL/AmpC-genes detected | R-gene on plasmid lost when subculturing<br>R-gene not included in database |
| | 2 | Resistance to third generation cephalosporins, but no ESBL-activity detected | No ESBL/Ampc-genes detected | Other resistance mechanism than ESBL-enzymes |
| **Sequence analysis-related** | 5 | Variable resistance | ESBL/AmpC-genes detected | Read depth <10x can suggest potential contamination of sequences or low copy plasmids |
| | 3 | Resistance | ESBL/AmpC-genes detected | Database included relevant genes, but lacked specific allele. Relevant hits missed when stringent cut-off was set (nucleotide identity and coverage 1) |
| | 13 | No resistance | ESBL/AmpC-genes detected | Increase of irrelevant hits when cut-off for nucleotide identity and coverage drops below 0.8 |

susceptible to all cephalosporins tested. This might be due to low expression of *bla* genes leading to an insufficient level of beta-lactamase activity to generate resistance ($bla_{CTX-M-3}$, $bla_{CTX-M-15}$ (n = 4)). Further four isolates with reduced susceptibility to third generation cephalosporins but a negative ESBL/AmpC-test. These isolates carried $bla_{DHA-1}$ (n = 2), $bla_{CTX-M-14}$ and $bla_{CTX-M-9}$, respectively. For seven isolates, ESBL/AmpC-production was demonstrated, but neither BLASTn nor srst2 reported any *bla* genes. These isolates may have had *bla* genes on mobile genetic elements that were lost during laboratory processing, or genes mediating beta-lactame-resistance that were not represented in the database. All isolates were stored in -80°C prior to sequencing which led to an average of four sub-cultivations prior to sequencing. In contrast, susceptibility testing was performed subsequently to isolation from the specimen; all cultures were handled within two days after they were grown. Finally, two isolates had reduced susceptibility to third generation cephalosporins, but had a negative clavulanic acid synergy test and no detected *bla* genes. For these two isolates, the observed reduced susceptibility might be due to other biological reasons.

Sequence analysis-related explanations to differences in phenotype and genotype were seen in five isolates where read depth for genetic targets dropped < 10x and varying results in phenotypic tests; ESBL/AmpC activity was demonstrated for three isolates, and two isolates were fully susceptible to cephalosporins. BLASTn reported *bla* genes ($bla_{CTX-M-15}$, $bla_{CTX-M-14}$) for two isolates, whereas *bla* genes for the remaining three isolates were only reported for an identity threshold 70% and for a maximum coverage of 60%, indicating incomplete assembly of the genes ($bla_{CTX-M\ 9-like}$, $bla_{CTX-M-14-like}$, $bla_{CTX-M-15-like}$). Three further isolates with phenotypic ESBL/AmpC-production, ESBL/AmpC genes were reported using both BLASTn and srst2 but only when cut-offs for nucleotide identity and coverage thresholds both set to 100%. The databases did contain relevant genes; however, correct alleles were lacking ($bla_{CTX-M-9-like}$, $bla_{CMY-15-like}$, $bla_{CTX-M-3-like}$). Finally, BLASTn searches with a coverage threshold of 40% in combination with an identity threshold of 70% resulted in hits for 13 isolates that were not producing ESBL/AmpC enzymes, nor were they reported by srst2. These reports were considered false positives due to low cut-off values. BLASTn reported $bla_{CMY-38}$ with 0.72 identity and 0.70 coverage for twelve isolates, and $bla_{CTX-M-42}$ with 1.0 identity and 0.41 coverage for one isolate. Read mapping with srst2 excluded all 13 isolates from the report, as the default minimal coverage was 0.9.

## Discussion

Genetic detection of resistance markers linked to ESBL/AmpC-production in *E. coli* using WGS yielded concordant results with phenotypic antimicrobial susceptibility testing for 91% of the included isolates. Similar results have been reported for *K. pneumoniae* using nanopore technology, with a reported concordance between 77% to 92% depending on the chosen data analysis procedures [21]. Likewise, Clausen *et al.* found an overall agreement of 95% to 96% between susceptibility testing and WGS analyses on a variety of species using different bioinformatical tools [22]. We found that many of these inconsistencies were due to underlying biology while others were due to cut-off dependencies in the bioinformatic analysis.

Incongruent results between phenotype and genotype can also have biological reasons, which is the potential explanation for isolates where ESBL/AmpC genes were detected but no resistance to third generation cephalosporins found. Silencing or low expression of antimicrobial resistance genes is not regularly considered in clinical settings and not necessarily revealed by routine susceptibility testing [9–11]. Since genetic methods like WGS can discover silenced genes, they can have an important complementary role in guiding effective antibiotic therapy, especially in situations of treatment failure. The synergy test for detection of production of ESBL/AmpC-enzymes was interpreted to be negative and therefore did not explain resistance to third generation cephalosporins. Although susceptibility testing is still able to provide guidance for anti-infective therapy, the resistance mechanism was clarified by WGS.

In some isolates, no ESBL/AmpC-genes were detected, despite phenotypic resistance to third generation cephalosporins and positive clavulanic acid synergy test. Potential explanations for this discrepancy could be that the isolates carried resistance genes that were not included in the database or that the plasmid was lost through subculturing prior to sequencing [23]. In two further isolates, neither ESBL/AmpC-genes nor ESBL/AmpC-enzyme activity was found, however resistance to third generation cephalosporins was seen. Both isolates were resistant to ceftazidime but susceptible to other third generation cephalosporins, which has previously been linked to alterations in outer membrane proteins [24].

Read depth is a parameter that is often used as a quality control parameter for reported genes and read depth below 10x has been regarded as likely caused by sequence contamination during the wet-lab process [22,25]. In this collection, five isolates had a read depth below 10x in common but for three of them ESBL/AmpC-production was confirmed. A potential biological explanation could be that ESBL/AmpC-genes can be carried by low-copy plasmids with significant expression [26]. However, for two isolates no phenotypic expression of ESBL/AmpC-enzymes was detected, making sequence contamination a plausible explanation. We therefore suggest not to automatically discard the finding of ESBL/AmpC-genes with <10x coverage, but instead report the genetic analysis as inconclusive and perform a supplementary phenotypic susceptibility testing.

When nucleotide identity dropped below 80% the list of reported isolates grew longer and consisted increasingly of irrelevant hits. In order to minimise false positive results with BLASTn, Zankari and Clausen *et al.* used 98% identity, and Doyle *et al.* recommended a sequence identity cut-off of at least 90% [15,22,27]. The nucleotide coverage was frequently set at 60% in order to increase retrieval of genes on the border of contigs [15,22]. In the present study, incomplete assembly of ESBL/AmpC genes was not found to be a significant problem, however, when the nucleotide coverage dropped below 80% a significant increase of the false discovery rate was seen while only one additional true finding was made.

From a technical point of view, the wet-lab part of WGS analysis has reached high interlaboratory reproducibility and accuracy [28]. In contrast, recommendations for bioinformatic parameter thresholds are of a more general character and should be specified for the antibiotic

and the resistance mechanism that the prediction targeted. The significance of the choice of thresholds and variables for the comparison of results from bioinformatic analyses has been shown in an interlaboratory comparison by Doyle *et al.* [15], who also stressed the need for an assessment of the results with respect to their clinical application. Hicks *et al* [29] showed that performance of WGS-based genotype-to-resistance-phenotype prediction varied by antimicrobial substance and bacterial species including its epidemiology.

Applications of WGS for resistance testing in clinical settings are not yet established and thus rarely used. Compared to PCR-based methods, WGS is still more time consuming and expensive, but it produces the basis for a more thorough genetic assessment of antimicrobial resistance. WGS has been discussed as a promising tool for rapid identification of resistance and several studies show promising results [21,30], phenotypic susceptibility testing remains the cornerstone for guiding antimicrobial treatment in acute and life-threatening infections. With rapid antimicrobial susceptibility testing available for clinically urgent situations, we believe that WGS-based methods are primarily useful as complementary method for the identification of causes for unclear phenotypic test results or treatment failure [31].

This study had limitations as it focused solely on the species *E. coli* and resistance to third generation cephalosporins and it is likely that other species and genetic resistance targets may lead to other results and further considerations. The number of isolates resistant to third generation cephalosporins is limited in comparison to susceptible isolates. Also, the study used a limited amount of bioinformatic tools: only one assembly algorithm was used along with a limited number of BLASTn parameters, and one read mapping program and it is possible that other algorithms might have given different results. Finally, quality criteria for WGS were set more moderately, compared to other studies, which may have resulted in an overestimation of the true positive rate.

## Conclusions

In conclusion, rapid advances in sequence technologies and phenotypic methods can complement present methods with detailed genetic background and information on growth behavior that is not covered by standard susceptibility testing which allows for further tailoring of antimicrobial chemotherapy. Yet, bioinformatical analysis of WGS data lacks specific guidelines for interpretation especially in a diagnostic meaning for clinical laboratories, and the present study underlined the need of consensus cut-offs in order to standardize pipelines. Furthermore, biological phenomena like heteroresistance, gene silencing, gene amplification or tolerance can cause incongruence between genotypic and phenotypic assays and have significant impact on the choice of the appropriate antibiotic therapy, emphasizing the need to further develop the repertoire of diagnostic methods.

## Acknowledgments

We would like to express our sincere gratitude to Professor Dan I. Andersson, Uppsala University, for his valuable input to improve this manuscript.

## Author Contributions

**Conceptualization:** Susanne Sütterlin.

**Data curation:** Anna Heydecke, Hong Yin.

**Formal analysis:** Anna Heydecke.

**Funding acquisition:** Susanne Sütterlin.

**Methodology:** Anna Heydecke, Hong Yin, Eva Tano, Susanne Sütterlin.

**Project administration:** Eva Tano.

**Supervision:** Eva Tano, Susanne Sütterlin.

**Validation:** Susanne Sütterlin.

**Writing – original draft:** Anna Heydecke.

**Writing – review & editing:** Hong Yin, Eva Tano, Susanne Sütterlin.

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
