## [Decision Letter · Decision Letter 0]

15 Sep 2023

PONE-D-23-01994Limitations in Predicting Reduced Susceptibility to Third Generation Cephalosporins in Escherichia coli Based on Whole Genome Sequence DataPLOS ONE

Dear Dr. Sütterlin,

Thank you for submitting your manuscript to PLOS ONE. After careful consideration, we feel that it has merit but does not fully meet PLOS ONE’s publication criteria as it currently stands. Therefore, we invite you to submit a revised version of the manuscript that addresses the points raised during the review process.

We look forward to receiving your revised manuscript.

Kind regards,

Mohamed O Ahmed, Ph.D

Academic Editor

PLOS ONE

Journal Requirements:

“SS holds a grant from ALF funds (ALF 810901) and the Swedish Research Council (2019-05909), and AH holds a grant from the Center for Research and Development Gävleborg, Uppsala University, Gävle, Sweden. Generation of sequence data was supported by a grant from Afa Insurance, Sweden (grant number 150411) and Alf-de-Ruvo memorial foundation, Sweden. The funders had no role in study design, data collection and analysis, decision to publish, or preparation of the manuscript.”

Reviewers' comments:

Reviewer's Responses to Questions

**Comments to the Author**

1. Is the manuscript technically sound, and do the data support the conclusions?

Reviewer #1: Yes

Reviewer #2: Yes

2. Has the statistical analysis been performed appropriately and rigorously? 

Reviewer #1: N/A

Reviewer #2: Yes

3. Have the authors made all data underlying the findings in their manuscript fully available?

Reviewer #1: Yes

Reviewer #2: Yes

4. Is the manuscript presented in an intelligible fashion and written in standard English?

Reviewer #1: No

Reviewer #2: Yes

5. Review Comments to the Author

Reviewer #1: The manuscript is technically sound and the data backs up the conclusion. However, there is no accession number for the data deposit. The manuscript is generally presented in a standard fashion. However, the writing grammar needs to be touched.

Authors explored the limitations in using whole genome sequence data to predict reduced susceptibility to third generation cephalosporins in Escherichia coli. It is of interest and could add benefits to the field.

The manuscript is straightforward and the results obtained support the discussion and conclusion. Results are presented and discussed clearly. However, the writing should be checked throughout. Some concerns are raised for manuscript revision.

1. Line 65, I think the use of conventional PCR is more common. Many labs do not have a realtime PCR machine.

2. Line 77, please correct “Escherichia coli” to “E. coli. Please also Please also check throughout.

3. Line 102, a reference for the ESBL screening is needed. Why cepodoxime and cefadroxil were used? For standard ESBL screening, cefotaxime and ceftazidime are mostly used as they are indicators for most ESBLs.

4. Line 207-209, Is there any supporting evidence for “This might be due to low expression……..(=3))”? May be due to mutations in the genes producing truncate proteins? Should “n” be equal to 4?

5. Line 212-213, please mention how many days the isolates were left at room temperature or how many passages to support the loss of bla genes on mobile genetic elements.

6. I agree with the note in Line 303-311.

7. Table 2, In the 2nd row of biology related, the potential explanation is not clear. Is it possible that the genes carried mutations?

8. There are misspellings, typo and grammatical errors throughout. Actually, I am not a native English speaker but I can tell that it is not for smooth reading. The writing should be touched for better reading. For example,

-Line 64, “betalactamases” should be corrected to “beta-lactamases”. Please also checkthroughout.

- Line 77, please correct “Escherichia coli” to “E. coli and also check throughout.

Etc.

Reviewer #2: The maniscript of Heydecke and colleagues adress discrepancies betwwen phenotypic and genotypic results applying to resistance to 3GC in E. coli. The paper is well written, the methology sounds well and the results are of interest, even if they are more confirmatory than really new. In particular, the authors clearly explained the reasons for discrepancies, either related to biological issues or to methodologies used for WGS data analysis.

General comment:

-the number of isolates tested (notably those resistant to 3GC is rather limited) and limits the scope of the results (this may be added in the limitations of the study

-Considering that WGS date are available, is there a link between discrepancies of phenotype and genotype and some specific clonal group?

Other comments

-L76: replace diagnose with detect

L212: this implies that phenotypic and genotypic analysis were not done simultaneously, is it correct?

L214: beta-lactam

Table 1: I think the added value of this table is weak and could be omitted

6. PLOS authors have the option to publish the peer review history of their article (what does this mean?). If published, this will include your full peer review and any attached files.

Reviewer #1: No

Reviewer #2: No

---

## [Author Response · Author response to Decision Letter 0]

2 Oct 2023

Response to reviewers PONE-D-23-01994

Limitations in Predicting Reduced Susceptibility to Third Generation Cephalosporins in Escherichia coli Based on Whole Genome Sequence Data

PLOS ONE

Dear Dr. Ahmed,

Please find enclosed our revised manuscript. Replies to each of the reviewers’ questions are given below.

We are very grateful for the constructive comments of the reviewers, which have helped us to improve our manuscript.

Yours sincerely

Susanne Sütterlin

Reviewer #1: The manuscript is technically sound and the data backs up the conclusion. However, there is no accession number for the data deposit. The manuscript is generally presented in a standard fashion. However, the writing grammar needs to be touched.

Authors explored the limitations in using whole genome sequence data to predict reduced susceptibility to third generation cephalosporins in Escherichia coli. It is of interest and could add benefits to the field.

The manuscript is straightforward and the results obtained support the discussion and conclusion. Results are presented and discussed clearly. However, the writing should be checked throughout. Some concerns are raised for manuscript revision.

1. Line 65, I think the use of conventional PCR is more common. Many labs do not have a realtime PCR machine.

In Sweden, the use of real-time PCR is standard, while conventional PCR is rarely used in diagnostics anymore. However, the article aims to be of interest to an international audience and thus it seems to be more appropriate to not specify the type of molecular method that can be used. We remove “real-time” and use only PCR.

2. Line 77, please correct “Escherichia coli” to “E. coli. Please also Please also check throughout.

We corrected accordingly and checked through the manuscript.

3. Line 102, a reference for the ESBL screening is needed. Why cepodoxime and cefadroxil were used? For standard ESBL screening, cefotaxime and ceftazidime are mostly used as they are indicators for most ESBLs.

We followed the recommendations from NordicAST (Nordic committee on antimicrobial susceptibility testing that represents all clinical laboratories in Scandinavia). NordicAST is represented in EUCAST, the European committee. While the use of cefpodoxime and cefadroxil has been removed from the European screening recommendations for ESBL production, the NordicAST kept them in their recommendations. 

(EUCAST guideline 2013: https://www.eucast.org/fileadmin/src/media/PDFs/EUCAST_files/Resistance_mechanisms/EUCAST_detection_of_resistance_mechanisms_v1.0_20131211.pdf

EUCAST guideline 2017: https://www.eucast.org/fileadmin/src/media/PDFs/EUCAST_files/Resistance_mechanisms/EUCAST_detection_of_resistance_mechanisms_170711.pdf

NordicAST recommendation 2021 (includes cefpodoxime): https://www.nordicast.org/d/6092?store_referer=true

NordicAST breakpoint table v13.0 (includes both cefpodoxime and cefadroxil): https://s3-eu-west-1.amazonaws.com/hl-intranet/files/a00a01520e275b6dfff4b1b7340cc06668471ef8/nordicast_bp_table_se_v_13_0_locked2.xlsx

Swedish clinical laboratories use cefadroxil commonly as screening disc in UTI samples, as it can also be used for treatment especially of lower UTI. When the isolate is susceptible to cefadroxil, ESBL production can be ruled out. It is a convenient and economic way of testing ESBL on standard agar plates with space for six antibiotic discs.

We would like to suggest to refer to both committee’s websites and use them as references. In order to clarify this, we have rewritten the according part of the manuscript.

4. Line 207-209, Is there any supporting evidence for “This might be due to low expression……..(=3))”? May be due to mutations in the genes producing truncate proteins? Should “n” be equal to 4?

Thank you for indicating our mistake, n is equal to 4, not 3, and has been corrected in the manuscript. We chose not to specify potential mechanisms leading to low expression of genes as we did not perform further analysis that can give further evidence for underlying mechanisms. However, we discuss the issue further in the discussion section.

5. Line 212-213, please mention how many days the isolates were left at room temperature or how many passages to support the loss of bla genes on mobile genetic elements.

That is a good point! We added information on amount of subculturing and leaving plates at room temperature in the manuscript.

6. I agree with the note in Line 303-311.

Ditto.

7. Table 2, In the 2nd row of biology related, the potential explanation is not clear. Is it possible that the genes carried mutations?

The isolates were resistant to cephalosporines, however, the synergy test and AmpC-test failed to reveal ESBL-production. Two isolates were AmpC-producers (DHA-1) that were not resistant to cefoxitin, and two carried CTX-9 and CTX-14), however, the synergy test did not indicate inhibition by beta-lactamase inhibitors (probably due to resistance to beta-lactamase inhibitors).

8. There are misspellings, typo and grammatical errors throughout. Actually, I am not a native English speaker but I can tell that it is not for smooth reading. The writing should be touched for better reading. For example,

-Line 64, “betalactamases” should be corrected to “beta-lactamases”. Please also checkthroughout.

- Line 77, please correct “Escherichia coli” to “E. coli and also check throughout.

Etc.

The manuscript was sent for language revision to a professional language editor and was checked thorough.

Reviewer #2: The maniscript of Heydecke and colleagues adress discrepancies betwwen phenotypic and genotypic results applying to resistance to 3GC in E. coli. The paper is well written, the methology sounds well and the results are of interest, even if they are more confirmatory than really new. In particular, the authors clearly explained the reasons for discrepancies, either related to biological issues or to methodologies used for WGS data analysis.

General comment:

-the number of isolates tested (notably those resistant to 3GC is rather limited) and limits the scope of the results (this may be added in the limitations of the study

We added this limitation to the according section of the manuscript.

-Considering that WGS date are available, is there a link between discrepancies of phenotype and genotype and some specific clonal group?

We have not analyzed the clonality of the isolates with regard to susceptibility systematically. However, as the isolates derive from different studies where clonality was investigated we did assess the possibility within the revision, but could not find conclusive patterns. 

Other comments

-L76: replace diagnose with detect

Changed.

L212: this implies that phenotypic and genotypic analysis were not done simultaneously, is it correct?

That is correct. As the first reviewer pointed out the same issue, we completed with a clarification accordingly.

L214: beta-lactam

Changed.

Table 1: I think the added value of this table is weak and could be omitted

After consideration we agree and omit the table accordingly.

---

## [Decision Letter · Decision Letter 1]

20 Nov 2023

Limitations in predicting reduced susceptibility to third generation cephalosporins in Escherichia coli based on whole genome sequence data

PONE-D-23-01994R1

Dear Dr. Sütterlin,

We’re pleased to inform you that your manuscript has been judged scientifically suitable for publication and will be formally accepted for publication once it meets all outstanding technical requirements.

Kind regards,

Mohamed O Ahmed, Ph.D

Academic Editor

PLOS ONE

---

## [Editor Report · Acceptance letter]

24 Nov 2023

PONE-D-23-01994R1 

Limitations in predicting reduced susceptibility to third generation cephalosporins in *Escherichia coli* based on whole genome sequence data 

Dear Dr. Sütterlin:

I'm pleased to inform you that your manuscript has been deemed suitable for publication in PLOS ONE. Congratulations! Your manuscript is now with our production department. 

Kind regards, 

on behalf of

Dr. Mohamed O Ahmed 

Academic Editor

PLOS ONE